 **eLIFE**

# Histone supply regulates S phase timing and cell cycle progression

**Ufuk Günesdogan[1,2], Herbert Jäckle[1], Alf Herzig[1,3]***

[1]Abteilung Molekulare Entwicklungsbiologie, Max-Planck-Institut für biophysikalische Chemie, Göttingen, Germany; [2]Wellcome Trust/Cancer Research UK Gurdon Institute, University of Cambridge, Cambridge, United Kingdom; [3]Abteilung Zelluläre Mikrobiologie, Max-Planck-Institut für Infektionsbiologie, Berlin, Germany

**Abstract** Eukaryotes package DNA into nucleosomes that contain a core of histone proteins. During DNA replication, nucleosomes are disrupted and re-assembled with newly synthesized histones and DNA. Despite much progress, it is still unclear why higher eukaryotes contain multiple core histone genes, how chromatin assembly is controlled, and how these processes are coordinated with cell cycle progression. We used a histone null mutation of *Drosophila melanogaster* to show that histone supply levels, provided by a defined number of transgenic histone genes, regulate the length of S phase during the cell cycle. Lack of de novo histone supply not only extends S phase, but also causes a cell cycle arrest during G2 phase, and thus prevents cells from entering mitosis. Our results suggest a novel cell cycle surveillance mechanism that monitors nucleosome assembly without involving the DNA repair pathways and exerts its effect via suppression of CDC25 phosphatase String expression.

***For correspondence:** herzig@ mpiib-berlin.mpg.de

**Reviewing editor**: Asifa Akhtar, Max Planck Institute for Immunobiology and Epigenetics, Germany

## Introduction

Chromatin assembly during DNA replication is crucial for the repackaging of newly synthesized DNA and for maintaining or erasing histone modifications. During this process, pre-existing or so-called parental histones are recycled and assembled into nucleosomes together with de novo synthesized histones (*Alabert and Groth, 2012*; *Annunziato, 2012*). To compensate for the high demand of histone proteins during DNA replication, the canonical histones H1, H2A, H2B, H3, and H4, which are encoded by multiple gene copies in higher eukaryotes, are highly and exclusively expressed in S phase of the cell cycle (*Marzluff et al., 2008*).

The assembly of chromatin is mediated by an interplay of components of the DNA replication machinery and histone chaperones, which mediate the deposition of histones into nucleosomes (*Alabert and Groth, 2012*; *Annunziato, 2012*). Apparently, the pace of DNA synthesis is tightly coupled to the assembly of newly synthesized DNA into chromatin. Multiple studies showed that the depletion of the histone chaperones Asf1 and CAF-1 results in a slow down of DNA synthesis during S phase (*Hoek and Stillman, 2003*; *Ye et al., 2003*; *Nabatiyan and Krude, 2004*; *Groth et al., 2007*; *Takami et al., 2007*) preceding the accumulation of DNA damage in mammalian cells (*Hoek and Stillman, 2003*; *Ye et al., 2003*). Also, diminishing histone supply during S phase through knock down of SLBP, which is required for histone mRNA stability and translation, decreases the rate of DNA synthesis (*Zhao et al., 2004*). A recent study that targeted SLBP together with FLASH, a factor that is required for histone mRNA transcription and processing (*Barcaroli et al., 2006*; *Yang et al., 2009*), revealed that replication fork progression depends on nucleosome assembly potentially through a mechanism based on a feedback from the histone chaperone CAF-1 to the replicative helicase and/or the unloading of PCNA from newly synthesized DNA upon nucleosome assembly (*Groth et al., 2007*; *Mejlvang et al., 2014*).

**eLife digest** As a cell prepares to divide, it goes through four distinct stages. First, it grows in size (G1 phase); next it copies its entire DNA content (S phase); then it grows some more (G2 phase); and, last, it splits into two new cells (M phase).

During S phase, groups of histone proteins that normally stick together to tightly package the DNA are pulled apart in order to make the DNA accessible for copying. After the DNA has been duplicated, both copies of the DNA strand need to be repackaged. Therefore, after copying the DNA the cell rapidly reassembles the DNA–histone complexes (called nucleosomes), using a combination of old and newly synthesized histones to do so. A cell can adjust how quickly it copies DNA according to the availability of these histone proteins, which is important because copying DNA without the resources to package it could expose the DNA to damage.

Here, Günesdogan et al. investigate how a cell controls these processes using a mutant of the fruit fly *Drosophila melanogaster* that completely lacks the genes required to make histones. Cells that lack histones copy their DNA very slowly but adding copies of histone genes back into these flies speeds up the rate at which DNA is copied.

Günesdogan et al. ask whether the slower speed of DNA replication in cells without new histones is connected to preventing DNA damage. However, these cells can still copy all their DNA, despite being unable to package it, so the higher risk of making mistakes is not enough to stop S phase. In fact, indications suggest that DNA damage detection methods continue to work as normal in cells without histones: these cells can get all the way to the end of G2 phase without any problems.

To go one step further and start splitting in two, a cell needs to switch on another gene, called *string* in the fruit fly and CDC25 in vertebrates, which makes an enzyme required for the cell division process. Normal cells switch on *string* during G2 phase, but cells that lack histones do not—and therefore do not enter M phase. Günesdogan et al. show that turning on *string* by a genetic trick is sufficient to overcome this cell cycle arrest and drive the cells into M phase. *String* could therefore form part of a surveillance mechanism that blocks cell division if DNA–histone complexes are not assembled correctly.

The coupling of replication fork progression and nucleosome assembly might compensate for short-term fluctuations in histone availability (*Mejlvang et al., 2014*). However, it is still unclear whether chromatin integrity is monitored after or during DNA replication. Genome integrity during S phase is governed by the ATR/Chk1 and ATM/Chk2 checkpoint mechanisms that sense replication stress and DNA damage, respectively (*Bartek and Lukas, 2007*; *Cimprich and Cortez, 2008*). Lack of CAF-1 or Asf1 function leads to accumulation of DNA damage and activation of the ATM/Chk2 pathway (*Hoek and Stillman, 2003*; *Ye et al., 2003*). These findings led to the hypothesis that chromatin assembly is monitored indirectly through accumulation of DNA lesions in response to stalled replication forks. However, since these chaperones have multiple functions such as unwinding of DNA during replication, in DNA repair (*Gaillard et al., 1996*; *Green and Almouzni, 2003*; *Schöpf et al., 2012*) as well as other nuclear processes (*Quivy et al., 2004*; *Houlard et al., 2006*). These multiple functions of these chaperones make it difficult to assess the direct effects of defective chromatin assembly.

Taking advantage of a histone null mutation in a higher eukaryote that recently became available in *Drosophila melanogaster* (*Günesdogan et al., 2010*), we directly addressed the requirement of canonical histone supply for DNA replication and cell cycle progression in a developing organism. By reintroducing a defined number of transgenic histone genes into the histone null mutant background, we show that the rate of DNA replication is coupled to the number of histone genes present in the genome and that histone supply is critical to coordinate S phase length with the developmental program. Surprisingly, cells that completely lacked de novo histone synthesis replicate DNA at a reduced rate, but complete S phase and arrest in cell cycle without accumulating DNA damage. This cell cycle arrest is mediated by suppressing the accumulation of transcripts encoding the CDC25 phosphatase String and provides evidence for a chromatin assembly surveillance mechanism that is independent of the known S phase checkpoints.

# Results

The histone null mutation in *D. melanogaster*, called *Df(2L)His$^C$*, lacks all genes encoding the canonical histones (*Günesdogan et al., 2010*). *Df(2L)His$^C$* homozygous mutant animals (hereafter referred to as *His$^C$* mutants) that are derived from heterozygous parents contain only maternal histone mRNA and proteins, which are sufficient to complete the first 14 cell division cycles of the embryo (*Günesdogan et al., 2010*). *His$^C$* mutant embryos arrest before the onset of mitosis in cycle 15 ($M_{15}$) (*Günesdogan et al., 2010*). This highly uniform phenotype is likely due to the degradation of maternal histone mRNAs during the first G2 phase of embryogenesis in cell cycle 14 (*Marzluff et al., 2008*; *O'Farrell et al., 1989*) combined with the complete lack of zygotic histone gene expression during S phase of cell cycle 15 ($S_{15}$) (*Günesdogan et al., 2010*). In order to verify that the lack of histone transcription also results in a diminished pool of histone proteins in $S_{15}$, we compared the protein levels of histone H2B and H3 of wild type embryos that are in $S_{15}$ at 4–5 hr after egg laying (AEL) to sorted *His$^C$* mutant embryos that are still in $S_{15}$ at 5.5–6.5 hr AEL (see below and *Günesdogan et al., 2010*) by quantitative Western blotting (*Figure 1A,B*). The approximate twofold reduction in the histone levels of *His$^C$* mutant embryos is consistent with the fact that these embryos lack synthesis of new histones in $S_{15}$ but still contain parental histones from chromatin that was assembled during cycle 14. To test whether the reduced supply of histones in *His$^C$* mutant embryos leads to a decrease in nucleosome formation, we carried out Micrococcal Nuclease (MNase) digestion assays on chromatin from sorted *His$^C$* mutant and wild type sibling embryos (*Figure 1C,D*, *Figure 1—figure supplement 1*). The results show that chromatin from *His$^C$* mutant embryos is more accessible to MNase than control chromatin, leading to a more rapid generation of mononucleosomal DNA fragments and reflecting a decrease in nucleosome occupancy in chromatin of the *His$^C$* mutants.

Arrested *His$^C$* mutant cells express high levels of mitotic Cyclin B suggesting a cell cycle arrest in G2 phase of cycle 15 ($G2_{15}$) before mitosis (*Günesdogan et al., 2010*; *Lehner and O'Farrell, 1990*; *Figure 2A*). Consistent with this, we did not observe degradation of the mitotic Cyclin A or assembly of mitotic spindles in *His$^C$* mutant cells (*Figure 2—figure supplement 1*). To assess DNA replication in $S_{15}$, we used BrdU incorporation assays to label newly synthesized DNA (*Figure 2*). The results show that most cells of *His$^C$* mutant embryos entered $S_{15}$, but the spatial pattern of replicating cells was different from the highly stereotyped wild type pattern (*Figure 2B–D*). In wild type embryos, about 5 hr AEL (*Figure 2C,E*), the ventral epidermal cells incorporated BrdU in $S_{15}$, whereas the lateral cells entered $G2_{15}$ and stopped BrdU incorporation. Dorsal epidermal cells had already passed through $M_{15}$ and incorporated BrdU in $S_{16}$. In contrast, the *His$^C$* mutant cells failed to reach $G2_{15}$ and the majority of the cells continued to incorporate BrdU at low levels in $S_{15}$ (*Figure 2D,F*). To test whether the extended S phase of *His$^C$* mutant cells is caused by a reduced rate of DNA synthesis, we shortened the BrdU labelling pulses from 15 to 5 min. In wild type cells, BrdU incorporation was detectable during $S_{15}$ (*Figure 2I–K*), whereas in *His$^C$* mutant cells BrdU incorporation was detected only in a few cells with low Cyclin B levels indicating that they were still in early $S_{15}$ (*Figure 2L–N*). Notably, in *His$^C$* mutants ventral cells showed a uniform pattern and dorsal cells a punctuate pattern of BrdU incorporation (*Figure 2G,H*), which is characteristic for the replication of euchromatin and heterochromatin in early and late S phase, respectively (*Shermoen et al., 2010*). Although we cannot exclude that the lack of histone synthesis in *His$^C$* mutants interfered with firing of individual origins, the results suggest that both early and late replication origins are activated in *His$^C$* mutants. In conclusion, the absence of de novo histone supply reduces the rate of DNA synthesis shortly after the initiation of DNA replication, resulting in an extended S phase in mutant cells.

It was previously shown that depletion of histones in cultured cells leads to a slow down of replication fork movement and it was proposed that this mechanism might avoid chromatin assembly defects due to short-term fluctuations in histone availability (*Mejlvang et al., 2014*). Thus, we asked whether S phase could be faithfully completed under diminished but constant histone supply and whether there is a direct dose dependent relation between histone synthesis and the length of S phase. We reintroduced defined numbers of histone gene units (*His-GUs*) into the genome of *His$^C$* mutant embryos. Embryos carrying either two (*2xHis-GU*) or six (*6xHis-GU*) units completed cell cycle 15 as shown by the degradation of Cyclin B in $M_{15}$ (*Figure 3A,B*). These embryos were not rescued and they died later towards the end of embryogenesis (*Figure 3—figure supplements 1 and 2*). Compared to wild type, the onset of $M_{15}$ was delayed in embryos with *2xHis-GUs* and *6xHis-GUs* by 2 hr and 1 hr, respectively, which was due to an extended S phase 15 (*Figure 3A–E*, *Figure 3—figure supplements 1 and 2*). In contrast,

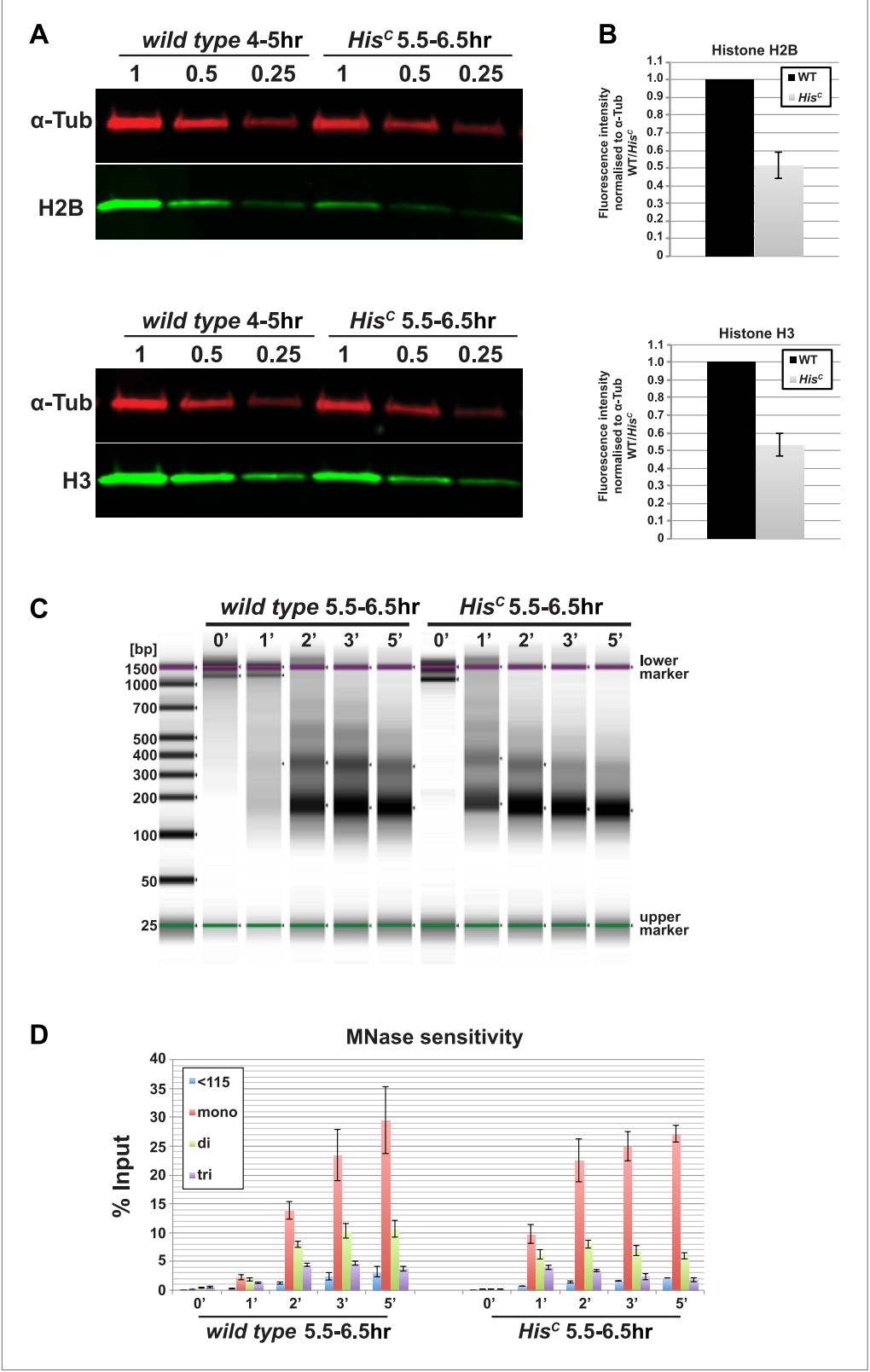

**Figure 1**. Nucleosome density is affected in *His^C* mutant cells. (**A**) Fluorescent Western blot analysis for histone H2B and H3 (green) using wild type embryos at 4–5 hr and sorted *His^C* mutant embryos at 5.5–6.5 hr AEL, respectively. α-Tubulin (α-Tub; red) was used as loading control. A dilution series of each extract was loaded (1, 0.5, 0.25).
*Figure 1. Continued on next page*

*Figure 1. Continued*

(**B**) Quantification of Western blots as shown in (**A**). Fluorescence measurements of the histone signal was normalised to the α-Tubulin signal and the wild type (WT)/*His^C* ratio is shown. The histone protein content is ~twofold reduced in *His^C* mutant embryos as compared to wild type. Mean values from three independent experiments are shown. Error bars indicate standard error. (**C**) Gel electrophoresis of time-course (0′–5′) micoccocal nuclease (MNase) digestions using sorted *His^C* mutant and wild type sibling embryos at 5.5–6.5 hr after egg laying, respectively. (**D**) Quantification of MNase digestion experiments as shown in (**C**). *His^C* mutant chromatin is digested more rapidly into mononucleosomal DNA than control chromatin. Mean values from three independent experiments are shown. Error bars indicate standard error.

The following figure supplement is available for figure 1:

**Figure supplement 1**. *His^C* mutant chromatin shows increased MNase sensitivity.

embryos containing *12xHis-GUs* were fully rescued and entered $M_{15}$ at the same time as the wild type cells, that is, 4.5–5 hr AEL as shown previously (**Günesdogan et al., 2010**). These results establish that the length of S phase directly correlates with the transgene-derived de novo histone supply. In addition, our data indicate that the mechanisms adjusting replication fork movement to the available histone supply allow the completion of S phase under conditions of permanently diminished new histone supply in vivo (**Figure 3F**).

Several studies showed that mammalian tissue culture cells depleted for CAF-1 accumulate in S phase due to a decreased rate of DNA replication (**Hoek and Stillman, 2003**; **Ye et al., 2003**; **Nabatiyan and Krude, 2004**; **Takami et al., 2007**) followed by accumulation of DNA damage and activation of the conventional DNA damage checkpoints (**Hoek and Stillman, 2003**; **Ye et al., 2003**). However, it remained unclear whether this S phase arrest represents a direct consequence of a failure in chromatin assembly as yeast cells, for example, can complete one round of replication after the depletion of histone H4 (**Kim et al., 1988**). In order to address whether DNA replication can be completed in the absence of de novo histone synthesis, we quantified the amount of nuclear DNA in DAPI-stained *His^C* mutant cells and compared it with wild type control cells during early $S_{16}$ (2N) and $G2_{15}$ (4N), respectively (**Figure 3G**). The DNA content of *His^C* mutant cells at 6.5–7.5 hr AEL corresponded to the 4N value of wild type nuclei in G2. This finding indicates that DNA replication in *His^C* mutant cells has essentially been completed.

To further explore whether *His^C* mutant cells accumulate DNA damage and activate the DNA damage checkpoints, we stained for the phosphorylated histone variant H2Av (γH2Av), which is like its vertebrate ortholog γH2AX a marker of DNA damage (**Madigan et al., 2002**). We found that γH2Av can be induced by ionizing irradiation that causes double strand breaks (DSBs) both in wild type and *His^C* mutant embryos, showing that the ATM/Chk2 checkpoint mechanism is still functional in *His^C* mutant cells (**Figure 4A,B**). We did not observe a difference with respect to γH2Av staining between non-irradiated *His^C* mutant and wild type cells in early $S_{15}$ (**Figure 4C,D**). However, we noted a slight increase in γH2Av staining between early and late $S_{15}$ in *His^C* mutant embryos (**Figure 4—figure supplement 1**). To further investigate this increase, we performed Western blot analysis for γH2Av. *Drosophila* S2R+ cells showed a dramatic increase in γH2Av upon treatment with hyrdoxyurea (HU) (**Figure 4E**, **Figure 4—figure supplement 1** and see below), which was not detectable in *His^C* mutant embryos undergoing late $S_{15}$ at 5.5–6.5 hr AEL when compared to wild type sibling embryos (**Figure 4E**).

To independently address the extend of DNA damage accumulation in *His^C* mutant embryos, we used TUNEL assays which support that *His^C* mutant cells do not accumulate significant levels of DNA damage until much later in development (≥10 hr AEL) when these embryos die (**Figure 4—figure supplement 2**). Finally, we performed genetic tests using mutants of *loki* (*lok*), the *Drosophila* DNA damage checkpoint kinase *chk2* (**Xu et al., 2001**). Homozygous *lok* mutants are viable and fertile. In contrast, *lok*, *His^C* double mutant embryos exhibited the *His^C* phenotype (**Figure 4F–H**). Hence, the cell cycle arrest in *His^C* mutant embryos is not mediated by the *chk2*-dependent DNA damage checkpoint pathway.

Mutations in the *Drosophila* ortholog of Chk1 (GRP) show developmental defects prior to cell cycle 15, excluding genetic experiments as we performed for *lok* (**Fogarty et al., 1994**; **Su et al., 1999**). Thus, we tested ATR/Chk1 checkpoint activation by using a phosphospecific antibody that recognizes the ATR-dependent phosphorylation of S345 in human Chk1 in response to replicative stress, for

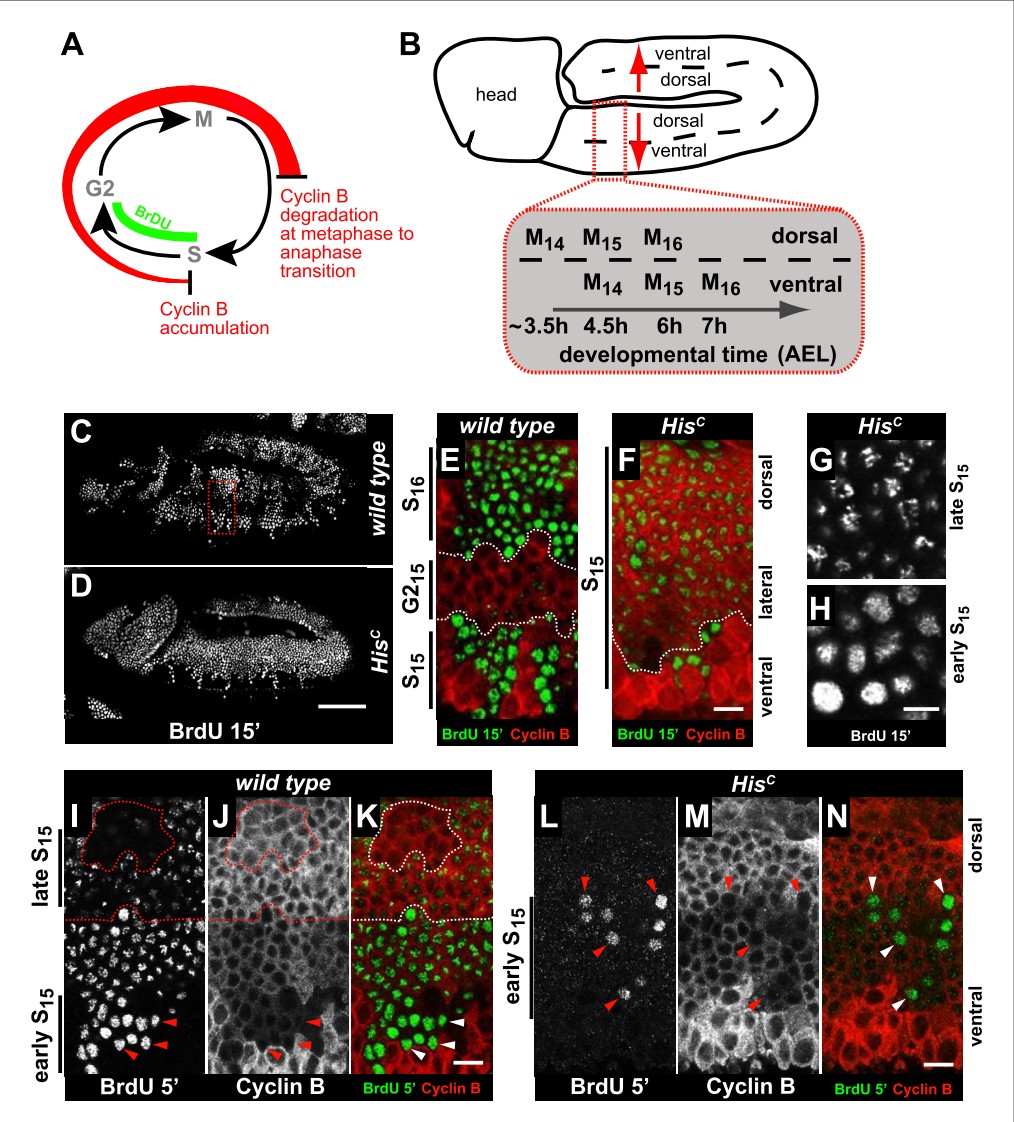

**Figure 2**. *His^C* mutant cells extend S phase and slow down DNA synthesis. (**A** and **B**) Schematic models of Cyclin B accumulation and BrdU incorporation during the embryonic cell cycle (**A**) and its spatial control (**B**). AEL: after egg laying. (**C–N**) BrdU pulse labelling for 15 (**C–H**) or 5 min (**I–N**) and staining with antibodies against BrdU (green in merge) and Cyclin B (red in merge). (**C** and **D**) BrdU was detected in the epidermis of *His^C* mutant embryos indicating DNA replication but the pattern of replicating cells was distinct from wild type. (**E** and **F**) Magnifications of an epidermal region as shown for the wild type embryo (boxed area in **C**). (**E**) BrdU labelled wild type cells of the ventral epidermis in S$_{15}$ (below dashed lines) and of the dorsal epidermis in S$_{16}$ (above dashed lines). Cells in G2$_{15}$ were BrdU negative with high levels of Cyclin B and located in the lateral epidermis (between dashed lines). (**F**) In *His^C* mutant embryos, lateral and dorsal cells re-accumulated Cyclin B and were labelled for BrdU (above dashed line), indicating that mutant cells still replicated DNA during S$_{15}$. (**G** and **H**) Punctate pattern of BrdU incorporation in cells that progressed into late S$_{15}$ in the dorsal epidermis (**G**) and uniform incorporation pattern in cells of the ventral epidermis (**H**). (**I–K**) In wild type embryos replicating cells in early (ventral, below dashed line) and late S$_{15}$ (dorsal, above dashed line) were detected after a 5 min BrdU labelling pulse. Some patches of dorsal cells completed S$_{15}$ and did not incorporate BrdU (encircled). (**L–N**) In *His^C* mutant embryos BrdU was detected after a 5 min BrdU labelling pulse only in ventral cells that were in early S phase as indicated by low levels of Cyclin B (arrowheads). Dorsal up (**E–N**), scale bars: 100 μm (**C** and **D**), 10 μm (**E–N**).

The following figure supplement is available for figure 2:

**Figure supplement 1**. The cell cycle arrest of *His^C* mutant cells is before mitosis.

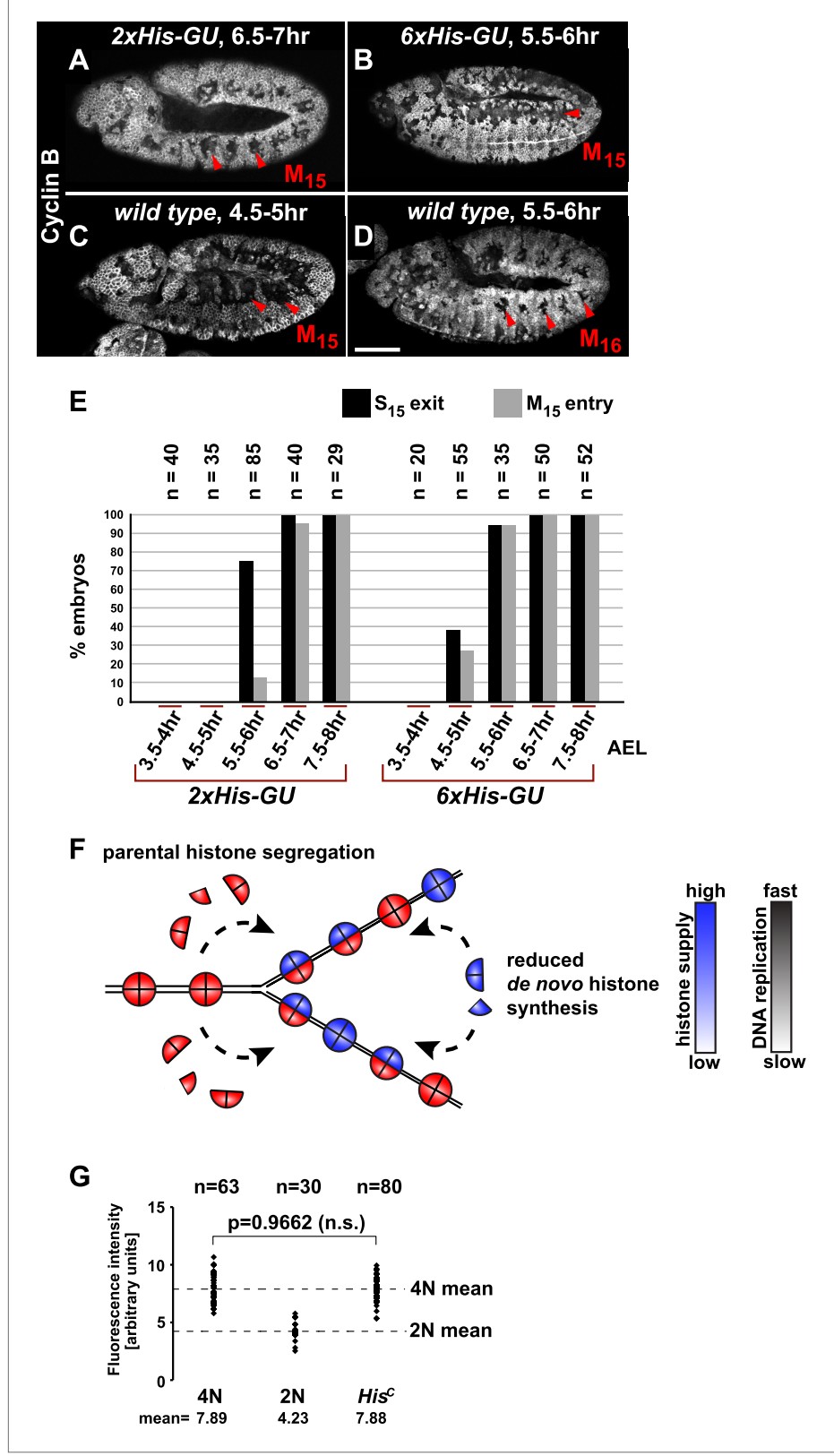

Figure 3. Histone availability determines the rate of S phase progression but is not required for completion of DNA replication. (A–D) Immunofluorescent staining with antibodies against Cyclin B. (A) Cyclin B degradation at 6.5–7 hr AEL around the tracheal pits in *2xHis-GU* embryos during $M_{15}$ (arrowheads) shows that these embryos are lagging
*Figure 3. Continued on next page*

*Figure 3. Continued*

more than one cell cycle behind as compared to wild type (see also *Figure 3—figure supplement 1*). (**B**) Cyclin B degradation in dorsal cells of *6xHis-GU* embryos at 5.5–6 hr AEL during $M_{15}$ (arrowheads), showing that *6xHis-GU* reduced the delay in cell cycle progression compared to *2xHis-GUs* (see also *Figure 3—figure supplement 2*). (**C**) Cyclin B degradation in dorsal cells of wild type embryos at 4.5–5 hr AEL during $M_{15}$ (arrowheads). (**D**) Cyclin B degradation at 5.5–6 hr AEL around the tracheal pits in wild type embryos during $M_{16}$ (arrowheads). (**E**) BrdU pulse labelling for 15 min of *2xHis-GU* and *6xHis-GU* embryos at indicated developmental stages and staining with antibodies against BrdU and Cyclin B. Shown is the quantification of embryos that completed $S_{15}$ ('$S_{15}$ exit', based on lack of BrdU labelling) and progressed into $M_{15}$ ('$M_{15}$ entry', based on Cyclin B degradation), showing the interdependence of the number of histone genes and cell cycle progression. n: number of embryos. (**F**) Schematic model showing nucleosome assembly at the replication fork and its dependence on histone supply. (**G**) DNA quantification of single nuclei stained with DAPI. Wild type nuclei in $G2_{15}$ and early $S_{16}$ defined 4N and 2N DNA content, respectively. *His*$^C$ mutant nuclei show mean intensity value of 4N nuclei, suggesting that mutant cells completed genome duplication. n: number of nuclei, p: probability from Student's *t* test, n.s.: not significant. Scale bars: 100 μm (**A**–**D**).

The following figure supplements are available for figure 3:

**Figure supplement 1**. Cell cycle progression of *His*$^C$ mutant embryos with two His-GUs (*2xHis-GUs*).

**Figure supplement 2**. Cell cycle progression of *His*$^C$ mutant embryos with six His-GUs (*6xHis-GUs*).

---

example, UV irradiation or HU treatment (*Zhao and Piwnica-Worms, 2001*). This antibody is expected to cross-react with *Drosophila* GRP due to sequence similarity and detected a single band in Western blots (*Figure 5—figure supplement 1*) as well as a clear signal in immunofluorescence (*Figure 5—figure supplement 2*) upon HU treatment of *Drosophila* S2R+ tissue culture cells. To test whether the ATR/Chk1 checkpoint is functional in *His*$^C$ mutant embryos, we irradiated embryos with UV light (254 nm, UVC), which induces replication stress and replication fork uncoupling (*Byun et al., 2005*; *Cimprich and Cortez, 2008*). Wild type embryos and *His*$^C$ mutant embryos accumulated phosphorylated GRP protein (pGRP) in response to UVC, showing that the checkpoint response is functional in the mutant embryos (*Figure 5A–D*). Without UVC treatment *His*$^C$ mutant embryos did not display elevated pGRP levels as compared to wild type (*Figure 5E–H*), which was also verified by Western blotting of extracts from sorted *His*$^C$ mutant and wild type sibling embryos (*Figure 5I*). In addition, treatment of *His*$^C$ mutant embryos with the ATR inhibitor VE-821 (*Prevo et al., 2012*) or the Chk1 inhibitor CHIR-124 (*Tse et al., 2007*) did not result in a release of the cell cycle arrest (*Figure 5—figure supplements 3 and 4*). In addition to DSBs, replicative stress can induce phosphorylation of H2Av (*Figure 4—figure supplement 1K–P*), either directly by ATR dependent phosphorylation (*Ward and Chen, 2001*; *Joyce et al., 2011*) or through interconversion of single-stranded DNA generated at stalled replication forks into DSBs (*Cimprich and Cortez, 2008*). Consistent with the notion that the DNA damage checkpoints are functional in *His*$^C$ mutant embryos we found accumulation of γH2Av in UVC-treated embryos to levels well above the background levels detected in untreated *His*$^C$ mutant embryos in late $S_{15}$ (*Figure 5—figure supplement 5*). Interestingly, UVC-treated embryos were able to enter $M_{15}$ while displaying levels of γH2Av comparable or above to what we observed in untreated *His*$^C$ mutant embryos (*Figure 5—figure supplement 6*). Taken together, these results strongly suggest that *His*$^C$ mutant cells complete DNA replication in S phase without inducing significant DNA damage or replication stress and that the cell cycle arrest at the G2/M transition in *His*$^C$ mutant cells is not mediated by the conventional S phase checkpoints.

Progression from the G2 phase into mitosis critically depends on the dephosphorylation and activation of Cyclin/Cdk complexes, which is accomplished by a single gene in *Drosophila* embryos, encoding the CDC25 phosphatase String (*Edgar and O'Farrell, 1990*). In wild type, *string* mRNA accumulates during G2 phase and becomes rapidly degraded after cells exit mitosis (*Edgar et al., 1994*). *string* transcription is highly dynamic, dictates the pattern of cell divisions during embryogenesis and is controlled by the activity of developmentally regulated transcription factors binding to *cis*-regulatory sequences spread over >30 kb of the *string* locus (*Edgar et al., 1994*). In contrast to wild type embryos which accumulate *string* mRNA in $G2_{15}$ cells of the dorsal epidermis, *His*$^C$ mutant embryos failed to accumulate *string* in the corresponding cells although they showed normal

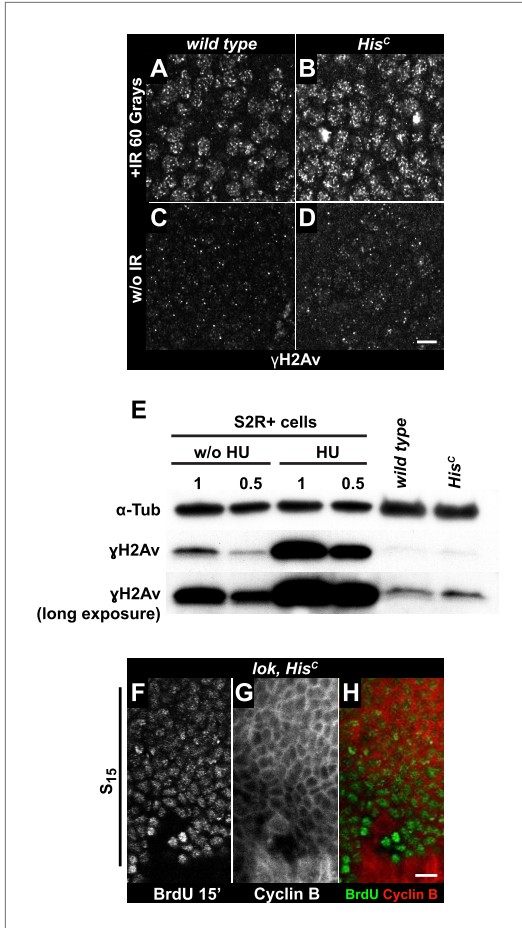

**Figure 4**. The cell cycle arrest of *His^C* mutant cells does not depend on the ATM/Chk2 DNA damage checkpoint. (**A** and **B**) Wild type and *His^C* mutant cells responded to ionizing irradiation (IR) by phosphorylation of the variant histone H2Av, detected by a phosphospecific antibody against H2Av (γH2Av). (**C** and **D**) Without irradiation, *His^C* mutant cells did not show elevated γH2Av staining compared to wild type, indicating that mutant cells did not accumulate DNA damage. (**E**) Western blot for γH2Av using untreated (w/o HU) and HU-treated (HU) S2R+ cells as controls (two dilutions, 1 and 0.5) as well as *His^C* mutant embryos and wild type embryos at 5.5–6.5 hr AEL. α-Tubulin (α-Tub) was used as a loading control. HU treatment results in a significant increase of γH2Av, which was not observed in *His^C* mutant embryos. (**F–H**) BrdU pulse labelling for 15 min and staining with antibodies against BrdU and Cyclin B. *lok, His^C* double mutant embryos showed a similar phenotype as *His^C* mutant embryos (see *Figure 1F*). Dorsal up in (**F–H**), scale bars: 10 µm.

The following figure supplements are available for figure 4:

**Figure supplement 1**. *His^C* mutant embryos show a moderate increase of γH2Av during $S_{15}$ progression.

*Figure 4. Continued on next page*

upregulation of *string* expression in ventral epidermal cells prior to $M_{14}$ (*Figure 6A,B*). Given the complex developmental regulation of *string*, we asked whether *string* transcription is disturbed in *His^C* mutant embryos due to misregulation of key patterning genes. However, we observed normal temporal and spatial expression patterns of developmental genes such as the segmentation gene *engrailed* and the homeotic genes *Ultrabithorax* and *Abdominal-B* in *His^C* mutant embryos (*Figure 6—figure supplement 1*). Hence, the developmental programme progresses normally in *His^C* mutant embryos up to the late embryonic stage when they eventually die (*Figure 4—figure supplement 2*). Similar to *His^C* mutant embryos, *string* mRNA was not detectable in dorsal epidermal cells of *2xHis-GU* embryos during their extended $S_{15}$ (*Figure 6C*). However, when wild type embryos undergo $M_{16}$ at 6.5–7 hr AEL (*Figure 6D,E*), *His^C* mutant embryos still failed to express *string* (*Figure 6F,G*) but *2xHis-GU* embryos upregulated *string* expression (*Figure 6H,I*). This result suggests that the failure of *His^C* mutant embryos to upregulate *string* after they finished replication in $S_{15}$ is not simply a consequence of their extended S phase but rather due to a surveillance mechanism that blocks the G2/M transition because chromatin assembly is not completed. It is interesting to note that the pattern of $M_{15}$ in *2xHis-GU* embryos closely resembled the pattern of $M_{16}$ in wild type embryos (*Günesdogan et al., 2010*; *Figure 6D,H*). This observation provides further support that the developmental programme of these embryos can progress normally, as *string* mRNA expression readjusts once DNA replication is completed. Thus, sufficient histone supply, as provided by multiple copies of *His-GUs*, is critical to the coordination of the developmental and cell division programmes during wild type embryogenesis of *Drosophila*. The results also explain why higher eukaryotes, which undergo rapid mitotic cell divisions during embryonic development, contain multiple *His-GUs* in their genomes.

If *string* activity is the only limiting factor that restricts cell cycle progression in *His^C* mutant cells, its ectopic expression should drive the G2 to M transition. To test this hypothesis, *His^C* mutant embryos were forced to express *string* in a striped pattern under the control of the *prd-GAL4* driver using the GAL4/UAS system (*Brand and Perrimon, 1993*), which was visualised by coexpression of a *UAS-EYFP* transgene. The *string* expressing *His^C* mutant cells entered $M_{15}$, degraded Cyclin B and reaccumulated Cyclin B after mitosis, whereas cells lacking *string* expression remained arrested (*Figure 6J–P*). Mitotic progression in wild type

*Figure 4. Continued*

**Figure supplement 2**. *His^C* mutant embryos do not show accumulation of DNA damage during early embryogenesis.

embryos follows a stereotyped pattern characterized by (i) Cyclin B degradation and sister chromatid separation at the metaphase to anaphase transition, (ii) spindle elongation at the transition from anaphase-a into anaphase-b, (iii) the onset of chromatin decondensation in telophase, and eventually by (iv) cytokinesis (*Figure 6Q*). The *string*-induced mitosis in *His^C* mutant embryos was normal up to the metaphase to anaphase transition. During anaphase-b and telophase, however, we observed lagging chromosomes or chromatin bridges in almost all of the cells (95.8%, n = 24). These bridges were eventually resolved during cytokinesis, and the cells entered into interphase of cell cycle 16 (*Figure 6Q*). Together, these data indicate that *string* transcription is indeed the limiting downstream factor that restricts cell cycle progression in the absence of de novo histone synthesis. In summary, our study shows that DNA replication and histone availability are tightly coupled. Lack of de novo histone synthesis causes a *string*-dependent cell cycle arrest in G2 phase, suggesting a novel chromatin assembly checkpoint monitoring chromatin integrity.

## Discussion

We used a recently generated null mutation for canonical histones to address the consequences of histone deprivation during metazoan development. In addition to canonical histones, eukaryotes express histone variants that can replace canonical histones in a specific genomic context (*Banaszynski et al., 2010*). Our results show that these histone variants do not compensate for the lack of canonical histone synthesis with regard to chromatin assembly and cell cycle progression. This could be due to insufficient expression of variant histones from their endogenous promoters as it has been shown for the variant histone H3.3, which can fully replace its canonical counterpart, histone H3, but only if it is expressed from within a histone gene unit like the canonical histone (*Hödl and Basler, 2012*). Alternatively, it could reflect structural divergence of the histone variants as in the case of His2Av (*van Daal et al., 1988*) and dBigH1 (*Perez-Montero et al., 2013*). It will be interesting to test whether individual histone mutations, like a mutation in H2B which does not have a variant histone in *Drosophila* (*Talbert et al., 2012*), will cause a similar cell cycle arrest as the histone null mutation *His^C*.

Our results provide evidence that canonical histone supply directly affects the rate of DNA synthesis (*Figure 2L–N*). This observation is in line with studies that targeted either histone chaperones (*Hoek and Stillman, 2003*; *Ye et al., 2003*; *Nabatiyan and Krude, 2004*; *Groth et al., 2007*; *Takami et al., 2007*) or histone mRNA through SLBP or FLASH (*Zhao et al., 2004*; *Barcaroli et al., 2006*; *Mejlvang et al., 2014*) to interfere with chromatin assembly in tissue culture cells. However, previous work on SLBP in multicellular organisms revealed pleiotropic effects (*Sullivan et al., 2001*; *Lanzotti et al., 2002*; *Pettitt et al., 2002*). Our data illustrate that an extension of the S phase duration caused by diminished histone supply allows a faithful completion of S phase and transition from G2 into M phase of the cell cycle. This S phase extension is likely to be caused by a direct effect of lowered histone availability on replication fork progression (*Groth et al., 2007*; *Mejlvang et al., 2014*) and not by a lack of origin firing, although we cannot exclude this possibility completely. It was previously shown that postblastodermal development in *Drosophila* embryos proceeds largely uncoupled from progression through cell cycles 14–16 (*Edgar et al., 1994*; *Meyer et al., 2002*). Therefore, histone availability limits S phase duration and appears to be a critical link between cell division and development.

In the absence of de novo histone synthesis, we find that cells arrest in G2 phase of the cell cycle without activating the known ATM/Chk2 and ATR/Chk1 checkpoints. This observation is in contrast to previous studies on CAF-1, which found that cells arrest in S phase and accumulate DNA damage (*Hoek and Stillman, 2003*; *Ye et al., 2003*). This discrepancy might in part be explained by the fact that histone chaperones also have a direct function in DNA repair (*Schöpf et al., 2012*); and thus, in the presence of an intact DNA repair/chromatin assembly machinery in *His^C* mutants, DNA is replicated without the accumulation of damage, even when histone supply is restricted to the parental load of histones. Alternatively, the accumulation of DNA damage in histone chaperone-depleted cells might be the consequence of a prolonged replication slow down, since it was shown that neither ATM/Chk2 nor ATR/Chk1 are activated as an immediate consequence of histone deprivation but only after prolonged incubation times (>48 hr) (*Mejlvang et al., 2014*). Based on our DNA quantification experiments, we found that the bulk of DNA replication in *His^C* mutants is completed by about 2 hr

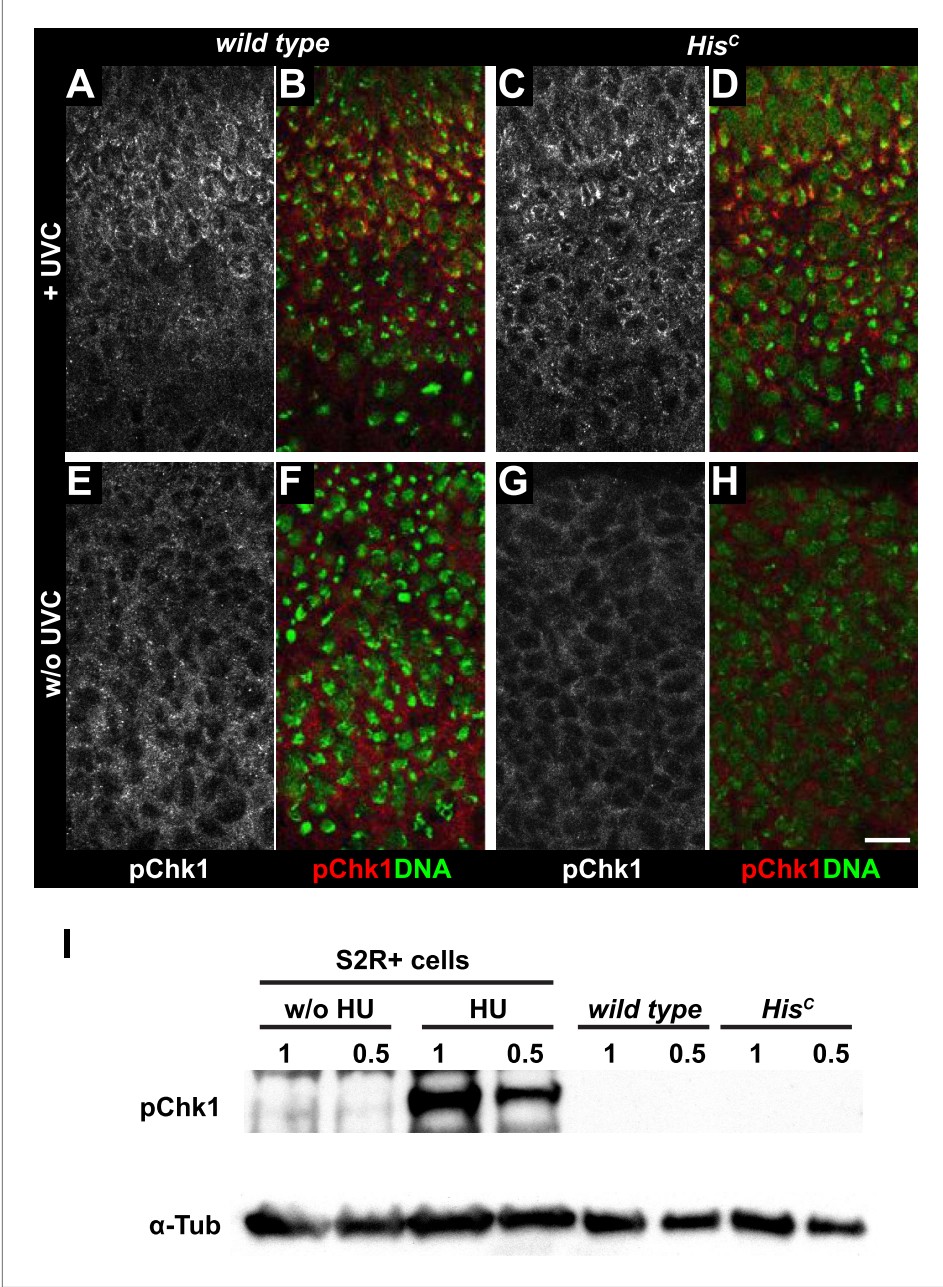

**Figure 5**. *His^C* mutant cells do not activate the ATR/Chk1 DNA damage checkpoint. (**A–D**) Wild type and *His^C* mutant cells responded to UV irradiation (UVC) by phosphorylation of the *Drosophila* Chk1 ortholog GRP, detected by a phosphospecific antibody for Chk1 (pChk1). Embryos were counterstained for DNA to visualize nuclei (**B** and **D**). (**E–H**) Without irradiation, *His^C* mutant cells did not show elevated staining for pChk1 compared to wild type, indicating that mutant cells did not activate the ATR/Chk1 checkpoint. (**I**) Western blot detecting pGRP by an antibody to phosphorylated Chk1 (pChk1) and α–Tubulin (α-Tub). Extracts were prepared from SR2+ tissue culture cells that were either untreated (w/o HU) or treated with HU (HU). Embryos were either sorted wild type controls or *His^C* mutants. Two dilutions (1 and 0.5) of each sample were loaded. Dorsal up in (**A–H**), scale bars: 10 μm.

The following figure supplements are available for figure 5:

**Figure supplement 1**. Detection of pGRP by a phosphospecific antibody for Chk1 by Western blotting.

**Figure supplement 2**. Detection of pGRP by a phosphospecific antibody for Chk1 by immunofluorescence.

*Figure 5. Continued on next page*

*Figure 5. Continued*

**Figure supplement 3**. Treatment of *His^C* mutant embryos with the ATR inhibitor VE-821 does not release the cell cycle arrest.

**Figure supplement 4**. Treatment of *His^C* mutant embryos with the Chk1 inhibitor CHIR-124 does not release the cell cycle arrest.

**Figure supplement 5**. UV irradiation induces phosphorylation of H2Av.

**Figure supplement 6**. Mitotic entry with elevated γH2Av levels.

after entry into S phase, which might differ from the timeframe required to develop significant DNA damage. Interestingly, we find that *His^C* mutant cells become TUNEL positive by about 6 hr after they enter S phase 15, which might reflect secondary DNA damage and/or cell death. Nevertheless, we found a moderate increase of γH2Av staining during late S phase in *His^C* mutant embryos. Our data indicate, however, that cells that resolved UVC-induced DNA damage, and therefore entered mitosis can do so with levels of γH2Av comparable to those we observe in *His^C* mutants. Thus, it is plausible that the slight increase in γH2Av in *His^C* mutants could result from incomplete turnover of γH2Av rather than directly reflect DNA damage that could activate the S phase checkpoints. Turnover of γH2Av was shown to require the Tip60 chromatin-remodelling complex (*Kusch et al., 2004*), which may be affected by the altered chromatin structure in *His^C* mutants. Alternatively, H2Av was recently shown to be phosphorylated independent of ATM/ATR by the chromosomal tandem kinase JIL-1 (*Jin et al., 1999*; *Thomas et al., 2014*), which may also be influenced by the changed chromatin topology in *His^C* mutants.

Both, the ATM/Chk2 and ATR/Chk1 checkpoints are known to act on CDC25 phosphatases by phosphorylation and protein destabilization (*Bartek and Lukas, 2007*) and it was shown in *Drosophila* that *string* transcripts accumulate normally in embryos that suffered from DNA damage (*Su et al., 2000*). In contrast, we find that *His^C* mutant cells fail to accumulate *string* transcripts when arrested in G2. This finding was surprising since it was shown that the temporal and spatial expression pattern of *string* is essentially unchanged in embryos that are arrested in G2 by mutations in *string* or in mitotic Cyclins (*Edgar et al., 1994*). Thus, this difference is likely due to the failure of *His^C* mutant embryos to assemble chromatin, resulting in a diminished nucleosome density as shown by the presence of excess MNase hypersensitive DNA. Although we cannot rule out that the lower abundance of histone proteins itself directly contributes to the G2 arrest, this possibility seems unlikely since histone levels rapidly decrease in G2 cells where the chromatin assembly surveillance should act (*Marzluff et al., 2008*). It remains unclear how the presence of unassembled chromatin is linked to the regulation of *string*, but the effect is specific, since *string* transcript accumulation is the only limiting factor to overcome the G2 arrest in *His^C*-mutant embryos. The subsequent mitosis in *His^C* mutants is completed and cells enter into the next cell cycle. Given that *His^C* mutant cells enter mitosis with presumably about half of the nucleosomes present in wild type chromatin, the mitotic defects like lagging anaphase chromosomes appear surprisingly mild. These defects could reflect problems in loading of structural components that are required for chromosome condensation and sister chromatid cohesion, like Cohesins and Condensins, which are proposed to require contact to chromatin rather than naked DNA (*Bernard et al., 2001*; *Nonaka et al., 2002*; *Tada et al., 2011*).

Taken together, our results suggest that incomplete chromatin assembly is monitored by a novel surveillance mechanism that can block cell cycle progression at the G2/M transition in *Drosophila*. Our findings now pave the way to address key questions regarding the orchestration of DNA synthesis and chromatin formation as well as the control of chromatin integrity during cell cycle progression.

## Materials and methods

### Fly strains

Construction of *Df(2L)His^C* and histone transgenes was described previously (*Günesdogan et al., 2010*). The *lok^P6 Df(2L)His^C* double mutant was constructed from *lok^P6* (*Abdu et al., 2002*) by meiotic

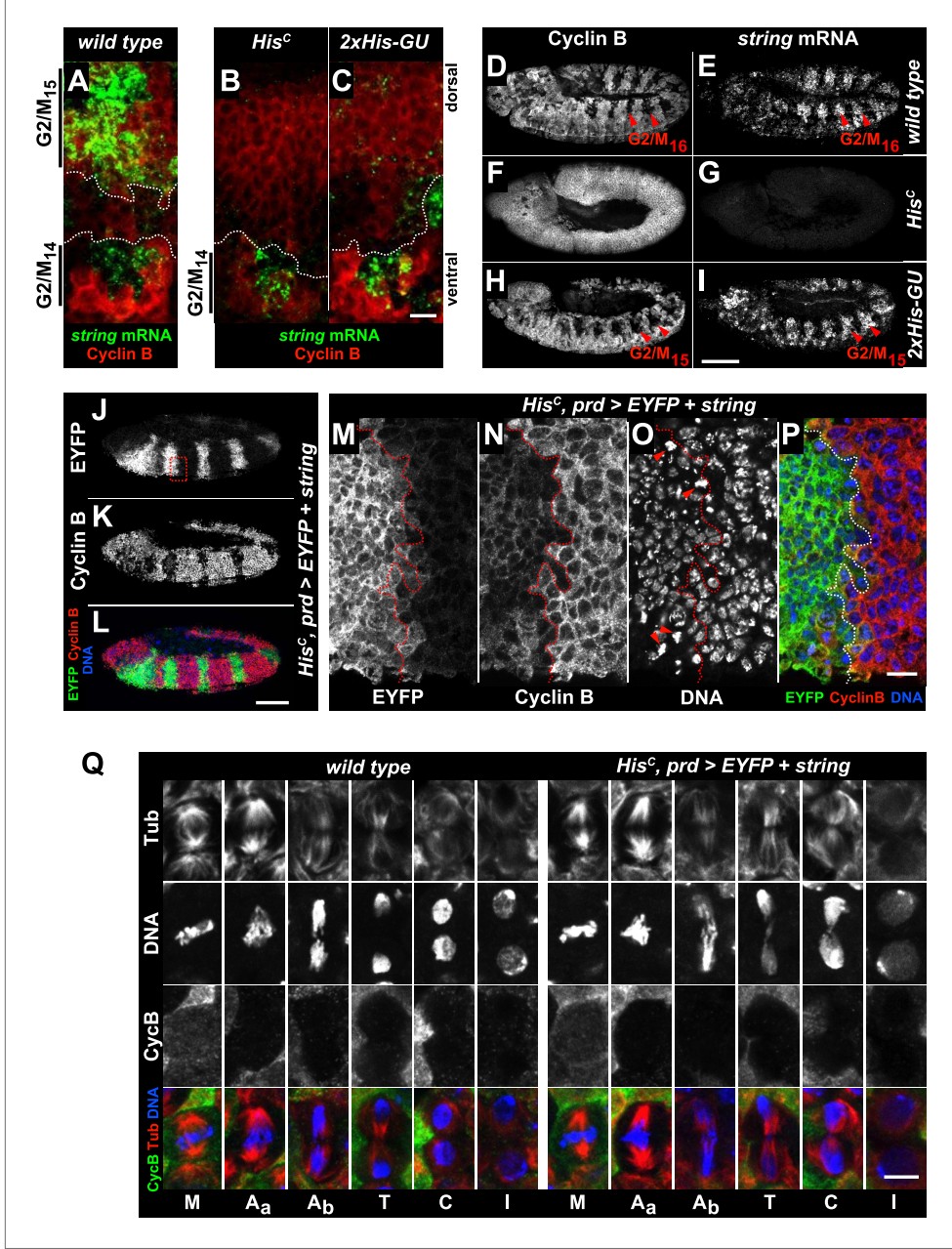

**Figure 6**. The cell cycle arrest of $His^C$ mutant cells depends on *string*. (**A–I**) *string* RNA in situ hybridisation (green in merge) and staining with an antibody against Cyclin B (red in merge). (**A–C**) Magnifications of epidermal cells from embryos at 4.5–5 hr AEL. (**A**) Wild type embryos upregulated *string* mRNA and accumulated Cyclin B in ventral epidermal cells during $G2_{14}$ (below dashed lines) and in dorsal epidermal cells during $G2_{15}$ (above dashed lines). In lateral cells with low levels of Cyclin B, *string* is not expressed during $S_{15}$ (between dashed lines). (**B**) $His^C$ mutant embryos failed to upregulate *string* in the dorsal epidermis (above dashed line). (**C**) *2xHis-GU* embryos failed to upregulate *string* in the dorsal epidermis (above dashed line). (**D–I**) Whole mount embryos at 6.5–7 hr AEL. (**D** and **E**) Dorsal cells of wild type embryos progressed into $G2_{16}$, upregulated *string* mRNA and degraded Cyclin B during $M_{16}$ (arrowheads). (**F** and **G**) $His^C$ mutant embryos did not degrade Cyclin B and failed to accumulate *string* mRNA. (**H** and **I**) *2xHis-GU* embryos accumulated *string* mRNA in $G2_{15}$ and degraded Cyclin B in $M_{15}$ (arrowheads). (**J–P**) $His^C$ mutant embryos expressing *UAS-EYFP* and *UAS-string* under the control of *prd-GAL4* stained with antibodies against Cyclin B and EYFP. (**M–P**) Magnifications of epidermal cells from an embryo (boxed in **J**). (**J–P**) Epidermal cells within the *EYFP*, *string* expression domains degraded Cyclin B and entered $M_{15}/S_{16}$. Arrowheads in (**O**) show mitotic cells. (**Q**) Mitotic progression in wild type $M_{15}$ and $M_{15}$ of $His^C$ mutant cells rescued by *string* expression visualized

*Figure 6. Continued on next page*

*Figure 6. Continued*

by staining for α-Tubulin (Tub), DNA, and Cyclin B (CycB). M: metaphase, A$_a$: anaphase-a, Ab: anaphase-b, T: telophase, C: cytokinesis, I: interphase. Dorsal up (**A–C**, **M–P**), scale bars 10 µm (**A–C**, **M–P**), 100 µm (**D–L**), 5 µm (**Q**).

The following figure supplement is available for figure 6:

**Figure supplement 1**. The expression pattern of developmental genes is normal in *His$^C$* mutant embryos.

recombination. In order to obtain maternal and zygotic *loki* mutant embryos, *lok$^{P6}$ Df(2L)His$^C$/lok$^{P6}$* animals were crossed *inter se*. For *string* expression, we constructed *Df(2L)His$^C$, P{UAS-2xEYFP}AH2* by meiotic recombination and generated embryos of the genotype *Df(2L)His$^C$, P{UAS-2xEYFP}AH2/ Df(2L)His$^C$, P{GAL4-prd.F}RG1/P{UAS-stg.N}4*. For sorting of mutant embryos, we constructed *Df(2L) His$^C$, P{GAL4-twi.2xPE}2* and selected EYFP-expressing embryos of the genotype *Df(2L)His$^C$, P{UAS-2xEYFP}AH2/Df(2L)His$^C$, P{GAL4-twi.2xPE}2*. In all other fly strains, *Df(2L)His$^C$* was heterozygous over the balancer chromosome *CyO, P{ftz/lacB}E3* to identify wild type sibling embryos. *6xHis-GU* flies were homozygous for the *M{3xHisGU.wt}ZH-86Fb* transgene. *2xHis-GU* flies were homozygous for the *M{1xHisGU.wt}ZH-86Fb* transgene (*Günesdogan et al., 2010*). *M{1xHisGU.wt}ZH-86Fb* was constructed analogous to *M{3xHisGU.wt}ZH-86Fb* except that pENTRL4R1-T1 and pENTRR2L3-T2 that both contained ~500 bp random DNA sequences instead of the His-GU were used for transgene construction.

## Embryo collections and staining

Time matched embryonic collections were obtained by restricting egg deposition to 30′ and subsequent aging of embryos at 25°C. Fixation, antibody staining, BrdU incorporation, and RNA in situ hybridisation procedures were described previously (*Günesdogan et al., 2010*). Different from our standard protocol, we reduced the fixation time to 5 min and used 37% PFA instead of 4% PFA to fix embryos after UVC treatment. For experiments that required visualization of the tubulin cytoskeleton, we fixed as described (*Karr and Alberts, 1986*). Tissue culture cells were grown in µ-slide eight-well ibidi dishes (IBIDI, Martinsried, Germany) and fixed and treated in these wells by the same procedures as embryos (*Günesdogan et al., 2010*). Primary antibodies used here were: rabbit anti-Cyclin B (*Jacobs et al., 1998*) (1: 3000), chicken anti-β-Galactosidase (1:1000; Abcam, Cambridge, UK), H2Av pS137 (γH2Av; 1:500; Rockland, Gilbertsville, PA), rabbit anti-Chk1 phospho-S345 (1:300; Abcam), mouse anti-α Tubulin DM1A (1:500; Sigma-Aldrich, Taufkirchen, Germany), chicken anti-Galactosidase (1:1000; Abcam), sheep anti-Digoxigenin (1:2000; Roche, Mannheim, Germany), and a mouse anti-BrdU antibody (1:80; Becton Dickinson, Heidelberg, Germany). Secondary antibodies were: goat anti-mouse IgG coupled to Alexa Fluor488 (1:400; Life Technologies, Paisley, UK), goat anti-rabbit IgG Alexa Fluor488 (1:400; Life Technologies), goat anti-rabbit IgG Alexa Fluor633 (1:400; Life Technologies), goat anti-rabbit IgG Alexa Fluor647 (1:400; Life Technologies), goat anti-chicken IgY Alexa 568 (1:400; Life Technologies), goat anti-chicken IgY Cy3 (1:400; Jackson ImmunoResearch, West Grove, PA), donkey anti-sheep IgG Biotin-SP (1:500; Jackson ImmunoResearch). DNA was stained with DRAQ-5 (Biostatus Limited, Shepshed, UK) or Vectashield DAPI (Vector Laboratories, Burlingame, CA). Antisense *string* RNA was obtained by in vitro transcription and detected as described previously (*Günesdogan et al., 2010*) using biotinylated secondary antibodies. For detection, embryos were incubated for 45′ with ABC reagents (Vector Laboratories), followed by a 5′ incubation with TSA Flourescein reagents (Perkin Elmer, Waltham, MA) diluted 1:50.

## Western blots

50–100 wild type or *His$^C$* mutant embryos were collected and lysed in 50 µl lysis buffer (50 mM Tris pH8, 150 mM NaCl, 0.5% Triton X-100, 1 mM MgCl$_2$, 0.1 mM EDTA and protease inhibitor [Roche]). Phosphatase inhibitor (Roche) was added for Western blots stained for pChk1 and γH2Av, respectively. 50 µl 4× Laemmli buffer was added and the samples were sonicated for 7 min (30 s interval, power 'low') using a Bioruptor (Diagenode, Liège, Belgium) and centrifuged for 10 min at 4°C at maximum speed. The supernatants were denatured at 95°C for 10 min before loading on BioRad 4–20% Mini-PROTEAN TGX gels. PVDF membranes (Merck Millipore, Billerica, MA) were used for blotting. After blotting, membranes were washed with TBS-T (Tris Buffered Saline with 0.1% Tween-20) and blocked for 1 hr in TBS-T and 5% dry milk powder. Primary antibodies diluted in blocking buffer were added

and incubated over night at 4°C. Membranes were washed 3× for 5′ in blocking buffer and secondary antibodies were added and incubated for 1 hr at room temperature. Membranes were washed 3× for 5′ in TBS-T. For fluorescent Western blots, images were acquired using an Odyssey infrared imaging system (LI-COR Biosciences, Lincoln, NE, USA). Quantifications were performed using the Image Studio Software (LI-COR Biosciences). For pChk1 and γH2Av Western blots, the ECL Western Blotting Kit (Pierce) was used to detect signals. Primary antibodies were: rabbit anti-Histone H3 (1:20000; Abcam), rabbit anti-Histone H2B (1:1000; Abcam), mouse anti-α Tubulin DM1A (1:2000; Sigma-Aldrich), rabbit anti-Chk1 phospho-S345 (1:1000; Abcam), and H2Av pS137 (γH2Av; 1:500; Rockland). Secondary antibodies were: goat anti-mouse IgG IRDye 680 (1:5000; LI-COR Biosciences), goat anti-rabbit IgG IRDye 800 (1:5000; LI-COR Biosciences), goat anti-rabbit IgG horseradish peroxidase conjugated, and goat anti-mouse IgG horseradish peroxidase conjugated (Sigma-Aldrich).

## Micrococcal nuclease assays

70–100 control or $His^C$ mutant embryos were collected and lysed in 50 µl micrococcal nuclease (MNase) digestion buffer (15 mM Tris, pH 7.5; 15 mM NaCl; 60 mM KCl; 0.34 M sucrose; 0.5 mM spermidine; 0.15 mM spermine; 1 mM PMSF; 0.5 mM DTT; 0.1% β-mercaptoethanol, protease inhibitor [Roche]). 200 µl MNase digestion buffer and 1 mM of $CaCl_2$ was added. The suspension was divided in equal aliquots. 500 gel units of MNase (NEB, Hitchin, UK) were added and the samples were incubated at 32°C in a heat block. The reaction was stopped by adding 5 µl of both 0.5 M EDTA and 0.5 M EGTA. 2.5 µl of 10% SDS and 1 µl Proteinase K (10 mg/ml) was added and incubated at 50°C over night. The DNA was purified using Ampure XP beads (Beckman Coulter, Brea, CA). The samples were analysed using an Agilent 2200 Tapestation system with High Sensitivity D1000 screen tapes (Agilent Technologies, Wokingham, UK). Genomic DNA screen tapes (Agilent Technologies) were used to determine the input (time point 0′) concentration. The Tapestation analysis software was used for quantification of following fractions: 60–115 bp (<115 bp), 115–270 bp (mononucleosomes), 270–455 bp (dinucleosomes), 455–650 bp (trinucleosomes).

## Irradiation of embryos, hydroxyurea treatment, treatment with inhibitors

Embryos were collected on apple-juice agar plates and aged to 4–5.5 hr AEL ($w^{1118}$, wild type) or 6–7 hr AEL ($His^C$ mutant embryos). This procedure yielded wild type embryos with cells in $G2_{15}$ and $His^C$ mutant embryos that were arrested in cell cycle progression. After irradiation at 60 Gray in a Torrex 150D (Astrophysics Research Corp., City of Industry, CA) embryos from both collections were aged for 20 min on the apple-juice agar plates and mixed before fixation. UV irradiation (254 nm) was done at 200 $mJ/cm^2$ as described (*Zhou and Steller, 2003*), and embryos were aged 45 min before fixation. $His^C$ mutant embryos were identified in the stained samples based on their cell division arrest phenotype.

For inhibitor treatment, we used the same procedure as for BrdU incorporation (*Günesdogan et al., 2010*), replacing the BrdU with 10 µM VE-821 (Selleckchem, Houston, TX) or 10 µM CHIR-124 (Selleckchem) and incubation of 45 min at RT before fixation.

S2R+ tissue culture cell were cultured in Schneiders Medium (Life Technologies), with 10% Fetal Calf Serum. Hydroxyurea (Sigma-Aldrich) was added to a final concentration 10mM and incubated for 12 hr. Extracts were prepared in SDS sample buffer after addition of phosphatase inhibitors (PhosSTOP, Roche) and protease inhibitors (cOmplete EDTA-free, Roche) at approximately $5 \times 10^8$ cells per ml.

## DNA quantification

Embryos were fixed by heat/methanol treatment and stained for Cyclin B. DNA was stained with DAPI (1:1000; Life Technologies). Cyclin B staining was used to distinguish homozygous mutant $Df(2L)His^C$ embryos and control embryos and to define the cell cycle stage of each cell. Stacks of nuclei were acquired with a 63× objective and a 10× optical zoom with a Leica TCS-SP5 AOBS confocal laser-scanning microscope (z-axis increment: 0.1 µm, 8 bit images, 512 × 512 pixel, 400 Hz scan speed). The gain and offset were adjusted once and then used for one complete experiment, avoiding saturation. Fluorescent measurements were carried out using ImageJ software. To define nuclear circumferences, we used the 'isodata thresholding' algorithm followed by manual inspection. This threshold we used in a customized macro (*Source code 1*) that utilizes the 'connected threshold grower' plugin of the ImageJ 3D toolkit to determine nuclear staining intensities in all slices of the z-stack. The absolute nuclear fluorescence intensities were calculated by integration of individual nuclear fluorescence

distributed over the image stack. For background detection five regions in between nuclei were analysed and their average was used for background subtraction.

### TUNEL assay

TUNEL assays were done as described (*Arama and Steller, 2006*) with some deviations. DIG-labelled nucleotides were detected with a sheep anti-DIG antibody (Roche) and a donkey biotinylated anti-sheep secondary antibody (Jackson ImmunoResearch). For signal amplification, embryos were incubated for 45′ with ABC reagents (Vector Laboratories), followed by a 5′ incubation with TSA Flourescein reagents (Perkin Elmer) diluted 1:50. Embryos were mounted in ProlongGold (Life Technologies).

## Acknowledgements

We thank Christian F Lehner for antibodies, comments, and suggestions; William Theurkauf for flies; Anja Schmidt for help with constructing fly stocks; Nick Brown for providing facilities; Benjamin Klapholz for technical advice; Lena Wartosch for help with Western blots. UG was supported by a fellowship from the Boehringer Ingelheim Fonds and a Marie Skłodowska-Curie fellowship. The work was supported by funds of the Max-Planck-Gesellschaft (HJ).

## Additional information

### Competing interests

HJ: Herbert Jäckle has been Vice President of the Max Planck Society—one of *eLife*'s three founding funders—and has attended until recently several meetings of eLife's board of directors. The other authors declare that no competing interests exist.

### Funding

| Funder | Grant reference number | Author |
| --- | --- | --- |
| Max-Planck-Gesellschaft | | Ufuk Günesdogan, Alf Herzig, Herbert Jäckle |
| Boehringer Ingelheim Fonds | Graduate Student Fellowship | Ufuk Günesdogan |
| European Commission | Marie Sklodowska-Curie fellowship | Ufuk Günesdogan |

The funders had no role in study design, data collection and interpretation, or the decision to submit the work for publication.

### Author contributions

UG, AH, Conception and design, Acquisition of data, Analysis and interpretation of data, Drafting or revising the article; HJ, Analysis and interpretation of data, Drafting or revising the article

## Additional files

### Supplementary file

• Source code 1. Source code of the ImageJ script used in this study to determine the volume and total fluorescence intensity of individual DAPI stained nuclei based on confocal z-stacks.

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
