## [Decision Letter]

Thank you for sending your work entitled “Histone supply regulates S phase timing and cell cycle progression” for consideration at *eLife*. Your article has been favorably evaluated by a Senior editor, Detlef Weigel, and 3 reviewers, one of whom is a member of our Board of Reviewing Editors.

The Reviewing editor and the other reviewers discussed their comments before we reached this decision, and the Reviewing editor has assembled the following comments to help you prepare a revised submission.

The reviewers had mixed opinions on the manuscript and its broader implication. After extensive discussion, we do appreciate the novel aspect of this study using *Drosophila* genetics to describe a possible new chromatin assembly surveillance that is mechanistically distinct from the ATR/Chk1 checkpoint. However, the reviewers felt that the manuscript needs to be further strengthened by adding additional experimental evidence in a revised version. In particular, it will be important to do the following:

1) DNaseI and/or MNase digestion experiments, which could reveal nucleosome density in HisC mutant cells.

2) Western blot analysis to assess the histone content of HisC mutant cells.

3) Assess nucleosome density in G2(15).

4) To provide further experimental evidence that G2 arrest is not dependent on ATR/Chk1.

5) Additional experiments strengthening String involvement are also highly encouraged.

In addition it is important that:

6) The published work is discussed more comprehensively.

7) The model is modified: there is no evidence that naked DNA at the forks is sending the signal, rather it is suggested to come from naked DNA in G2 cells.

8) It is possible to test whether treatment with chk1 and ATR inhibitors would not be feasible in flies? There are specific inhibitors available, which are widely used in the damage field.

---

## [Author Response]

*1) DNaseI and/or MNase digestion experiments, which could reveal nucleosome density in HisC mutant cells*.

As requested, we performed time-course micrococcal nuclease (MNase) digestion experiments with chromatin isolated from sorted *His*^*C*^ mutant and from wild type (sibling) embryos, respectively. We quantified the amount of mono-, di- and trinucleosomes for each time point and found that *His*^*C*^ mutant chromatin is digested significantly more efficient than control chromatin, resulting in a rapid generation of mononucleosomal DNA. Thus, these results indicate that nucleosome density is indeed affected in *His*^*C*^ mutant cells as reflected in an increased MNase sensitivity. We added this data into an extra figure (Figure 1) in the text and as a supplementary figure (Figure 1—figure supplement 1).

*2) Western blot analysis to assess the histone content of HisC mutant cells*.

We showed previously that *His*^*C*^ mutant embryos completely lack zygotic transcription of histone mRNA (18). Thus, only the pre-existing (parental) load of nucleosomes is available for chromatin assembly during S phase 15 in HisC mutant embryos, when in wild type zygotic transcription of histone genes commences. To show this, we performed quantitative Western blotting for histone H2B and H3, respectively, using protein lysates from sorted *His*^*C*^ mutant and wild type embryos. The results show that the histone protein content is about two-fold reduced in *His*^*C*^ mutant embryos. This result confirms that only parental histones are present in mutant embryos. We added this data into an extra figure (Figure 1).

*3) Assess nucleosome density in G2(15)*.

As described above, we performed MNase experiments and Western blots to determine nucleosome density and histone protein content.

*4) To provide further experimental evidence that G2 arrest is not dependent on ATR/Chk1*.

Mutations in *grapes* (*grp*), encoding the *Drosophila* ortholog of Chk1, cause developmental defects before cell cycle 15. Thus, one cannot perform experiments to test the involvement of ATR/Chk1 in the cell cycle arrest observed in *His*^*C*^ mutant embryos by genetic means. As an alternative approach, we performed an immunofluorescence analysis using a phosphospecific antibody that detects phosphorylated and activated GRP protein. Irradiation of *His*^*C*^ mutant embryos and wild type sibling embryos with UV light, known to induce stress-dependent replication effects, showed detectable levels of phosphorylated GRP. Thus, the ATR/Chk1 checkpoint must be still functional in *His*^*C*^ mutant embryos. However, untreated *His*^*C*^ mutant embryos did not exhibit detectable levels of pGRP, which we confirmed by Western blot analysis. As suggested by the reviewers, we also treated *His*^*C*^ mutant embryos with an ATR inhibitor (VE-821) and Chk1 inhibitor (CHIR-124), respectively, which did not result in a release of the cell cycle arrest in *His*^*C*^ mutant embryos. Taken together, these results strongly suggest that the lack of *de novo* histone supply does not induce replication stress thereby inducing the ATR/Chk1 checkpoint. We added these data in a new Figure 5 of the manuscript and in Figure 5—figure supplement 1, Figure 5—figure supplement 2, Figure 5—figure supplement 3, Figure 5—figure supplement 4, Figure 5—figure supplement 5 and Figure 5—figure supplement 6.

*5) Additional experiments strengthening String involvement are also highly encouraged*.

We show that *string* expression is sufficient to overcome the cell cycle arrest in *His*^*C*^ mutant embryos. If *string* expression would lead to the entry into mitosis, with massively unrepaired or under-replicated DNA, thereby overwriting the still functional DNA damage checkpoints, severe defects in mitosis would be expected (that may even lead to a so called ‘mitotic catastrophe’). We have now carefully documented the mitotic progression in *His*^*C*^ mutant cells upon expression of *string* and we found only minor mitotic defects. These data strengthen the involvement of *string*. They support the argument that it is indeed the lack of *string* transcript accumulation (rather than unresolved DNA damage) that is responsible for the cell cycle arrest in *His*^*C*^ mutant cells. We added these results to Figure 6.

*In addition it is important that*:

*6) The published work is discussed more comprehensively*.

We improved the revised version of the manuscript by adding additional data and discussed our study in the context of published work more comprehensively. We have also separated the Results and Discussion sections.

*7) The model is modified: there is no evidence that naked DNA at the forks is sending the signal, rather it is suggested to come from naked DNA in G2 cells*.

We took this model out of our manuscript.

*8) It is possible to test whether treatment with chk1 and ATR inhibitors would not be feasible in flies? There are specific inhibitors available, which are widely used in the damage field*.

We thank the reviewers for this suggestion. As described above, we performed experiments using an ATR and Chk1 inhibitor, respectively. Our results clearly support the argument that ATR/Chk1 is not involved in the cell cycle arrest observed in *His*^*C*^ mutant embryos (see details above).